# Maternal mid-pregnancy dietary patterns and inflammatory bowel disease in offspring from a prospective cohort study

Olivia Mariella Anneberg [1] ✉, Sjurdur Frodi Olsen[2,3], Anne Vinkel Hansen [1], Mette Julsgaard[1,4,5], Anne Ahrendt Bjerregaard[2,6], Thorhallur Ingi Halldorsson[2,7], Tine Jess [1,8] & Maiara Brusco De Freitas[1]

## Abstract

**Background:** Maternal diet during pregnancy may shape the child's immune system and gut microbiota maturation, potentially influencing the risk of developing inflammatory bowel disease (IBD). Therefore, we examined maternal a posteriori dietary patterns during pregnancy and their associations with pediatric-onset IBD risk in offspring.
**Methods:** The study comprised mother-child pairs from the Danish National Birth Cohort, which is a nationwide cohort of Danish pregnant mothers and their children, enrolled in 1996-2002 and followed prospectively through national health registers. Maternal dietary patterns during pregnancy were identified using k-means cluster analysis of 37 food group intakes, assessed with a food frequency questionnaire in gestational week 25 (second trimester). Pediatric-onset IBD cases (diagnosed at ≤18 years) were identified in Danish health registers. Cox regression explored associations between maternal dietary patterns during pregnancy and risk of pediatric-onset IBD in offspring, using the most common dietary pattern as reference.
**Results:** Based on analysis of 56,097 mother-child pairs, of which 161 (0.29%) offspring developed pediatric-onset IBD, we identify five distinct maternal dietary patterns: diverse (n = 8388), non-recommended (n = 14,110), Mediterranean (n = 14,700), animal-based (n = 3808), and Western (n = 15,091). Notably, a diverse dietary pattern, characterized by high intakes of most food groups, is associated with 45% lower risk of pediatric-onset IBD in offspring compared to a Western pattern (hazard ratio: 0.55; 95% confidence interval: 0.31–0.97). No significant associations are observed for the other patterns.
**Conclusions:** This large prospective cohort study shows that maternal adherence to a diverse dietary pattern during pregnancy may contribute to lower risk of pediatric-onset IBD in offspring.

## Plain language summary

This study investigates whether mothers' diets during pregnancy are linked to the risk of their children developing inflammatory bowel disease (IBD). We analyzed the diets of more than 56,000 pregnant mothers from Denmark and grouped them based on similarities in their eating patterns. Then, we retrieved medical information from national health registers about their children to identify who developed IBD. We found that children whose mothers followed a diverse diet during pregnancy, characterized by high intakes of various food groups (meat, fish, vegetables, legumes, fruits, potatoes, desserts), had lower risk of developing IBD than children of mothers with a typical Western-style diet. These results suggest that eating a diverse diet during pregnancy could have lasting benefits for children's health.

Inflammatory bowel disease (IBD), including Crohn's disease (CD) and ulcerative colitis (UC), is estimated to affect around 7 million people worldwide[1]. The disease typically begins in early adulthood, but recently, an increasing number of children are being diagnosed[2]. This concerning trend toward earlier disease onset underscores the need for early

prevention, as IBD with pediatric onset is often more severe[3] and may result in linear growth failure, delayed puberty, and impaired psychosocial functioning[4,5]. Meanwhile, the prevalence of both pediatric- and adult-onset IBD has steadily been rising on a global perspective, particularly in newly industrialized countries[2,6], suggesting that a Western lifestyle

[1]Center for Molecular Prediction of Inflammatory Bowel Disease – PREDICT, Department of Clinical Medicine, Aalborg University, Copenhagen, Denmark. [2]Department of Epidemiology Research, Statens Serum Institut, Copenhagen, Denmark. [3]Department of Nutrition, Harvard TH Chan School of Public Health, Boston, MA, USA. [4]Department of Hepatology and Gastroenterology, Aarhus University Hospital, Aarhus, Denmark. [5]Department of Clinical Medicine, Aarhus University, Aarhus, Denmark. [6]Center for Clinical Research and Prevention, Bispebjerg and Frederiksberg Hospital, Frederiksberg, Denmark. [7]Faculty of Food Science and Nutrition, School of Health Sciences, University of Iceland, Reykjavik, Iceland. [8]Department of Gastroenterology and Hepatology, Aalborg University Hospital, Aalborg, Denmark. ✉e-mail: omra@dcm.aau.dk

contributes to the disease development[7,8], possibly with influence already in early life[9].

According to the Developmental Origins of Health and Disease theory, early life exposures have a significant impact on long-term health[10,11]. This concept was first proposed by Barker et al. in 1989 after observing a link between low birth weight and death from coronary heart disease[12], which led to the hypothesis that in-utero undernutrition affects disease susceptibility later in life through its influence on fetal growth[13]. Since then, a growing body of evidence suggests a role of maternal diet during pregnancy in offspring's disease development, including cardiovascular disease, cancer, and allergies[14,15]. Likewise, maternal diet during pregnancy may contribute to offspring's IBD risk by inducing pathological changes similar to those observed in the preclinical phase of the disease[16], as shown by experimental studies with pregnant mice and their offspring[17]. For instance, maternal high-fat diets have consistently been found to increase offspring's intestinal inflammation and permeability, with release of pro-inflammatory cytokines, gut microbiota dysbiosis, and enhanced susceptibility to experimental colitis[18–21]. In contrast, the opposite effects have been observed with maternal high-fiber diets[22–24]. Furthermore, a high-sugar/high-fat diet, mimicking a classical Western diet, increased offspring's susceptibility to CD-like colitis characterized by severe intestinal inflammation and higher histopathological scores[25]. However, human studies confirming these findings are scarce[26,27].

Because the human diet is inherently more complex, focusing only on individual nutrients or foods has some conceptual limitations. People do not consume nutrients in isolation, but rather meals composed of different foods with diverse nutritional profiles, which may interact and influence health outcomes synergistically[28]. Dietary pattern analysis addresses this complexity by capturing overall intake, thereby accounting for the collinearity between dietary components and providing a more realistic depiction of habitual eating patterns[28]. Several dietary pattern analysis methods exist, usually divided into a priori (hypothesis-driven), a posteriori (data-driven), and hybrid approaches. Cluster analysis is an a posteriori approach that empirically derives dietary patterns by separating the study population into clusters based on similarities in their diets[29]. Thus, naturally occurring dietary patterns in the data can be identified and compared regarding offspring's disease risk, which may lay the foundation for future research exploring the specific food groups that drive these associations.

Therefore, the present study aim to characterize maternal a posteriori dietary patterns during pregnancy using k-means cluster analysis and to explore their associations with pediatric-onset IBD risk in the offspring. Utilizing data from the unique large-scale nationwide Danish National Birth Cohort (DNBC), we link detailed maternal dietary information collected in the second trimester of pregnancy with long-term data on their children obtained from the Danish National Patient Registry. Among the 56,097 mother-child pairs analyzed, we identify five distinct maternal dietary patterns during pregnancy: diverse, non-recommended, Mediterranean, animal-based, and Western. In this study, we show that compared to a Western dietary pattern, maternal adherence to a diverse dietary pattern during pregnancy is associated with a lower risk of pediatric-onset IBD in offspring, whereas no significant associations are observed for the remaining dietary patterns.

## Methods
### Study design and population
This prospective cohort study encompasses data from the DNBC, an ongoing study of early life exposures and their long-term effects on disease susceptibility[30]. Between January 1996 and October 2002, general practitioners across Denmark invited mothers to enroll in the DNBC at their first pregnancy visit, which usually takes place around gestational weeks 6 to 12[30]. All Danish-speaking mothers in Denmark, who became pregnant during the recruitment period and wished to carry to term, were eligible for inclusion[30].

Methods for data collection in the DNBC have been previously described[30,31]. In short, information on maternal and offspring exposures was collected through computer-assisted telephone interviews in gestational weeks 12 and 30, as well as 6 and 18 months post-delivery. In addition, a validated semi-quantitative food frequency questionnaire was mailed to the mothers in gestational week 25 to assess their diets during the previous four weeks[32,33]. This time frame was chosen because maternal dietary habits are more stable in the second trimester compared to earlier in the pregnancy period[31].

By October 2002, the DNBC had enrolled 101,042 mothers, corresponding to approximately 60% of those who were invited[30]. The enrolled mothers and their children were then followed long-term by linkage with the Danish National Patient Registry through their civil registration numbers. Our study included all singleton mother-child pairs from the DNBC, where the mother fully responded to the food frequency questionnaire with realistic self-reported energy intakes (>2500 kJ/day to <25,000 kJ/day). Mother-child pairs, where the child's information was missing from the patient registry, were excluded.

### Dietary pattern analysis
The food frequency questionnaire assessed the maternal frequency of consumption for ~360 food items. Reported frequencies were converted into times/day and multiplied by standard portion sizes to estimate daily food intakes (servings/day)[31]. Energy intake was previously calculated by combining estimated food intakes with national food composition tables[34], accounting for loss of fat, water, vitamins, and minerals during food preparation[31]. Furthermore, food intakes were translated into the intake of 37 food groups using standard classifications[34] (Supplementary Table 1) and scaled by z-score normalization to make them comparable.

Maternal dietary patterns derived from the scaled food group intakes with k-means cluster analysis using the R-package 'cluster'[35]. This method grouped mother-child pairs into $n$ mutually exclusive clusters based on maternal intake of the 37 food groups, where $n =$ the optimal number of clusters. The optimal number of clusters was determined using the Elbow method, which identified the point where additional clusters no longer improved the within-cluster homogeneity[36]. Clusters were internally validated by re-running the algorithm with a different number of initial seeds, showing consistently produced stable cluster centroids.

Thereby, each mother-child pair was assigned to a single cluster based on maternal dietary patterns. For each cluster, the mean scaled food group intakes (cluster centroids) were visualized in a spider plot to identify and name the dietary pattern that defined the mothers belonging to the specific cluster. Dietary patterns were named based on the predominant food groups consumed within each cluster, aiming to use terminology consistent with previous studies to facilitate comparison and communication of the results[37–39].

### Outcome
Children with pediatric-onset IBD, defined as ≤18 years of age at the time of their diagnosis[40], were identified by linkage with medical data in the National Patient Registry through their civil registration numbers. Positive cases were defined as at least two IBD hospital contacts within two years, consisting of inpatient contacts only or a combination of inpatient and outpatient contacts[41]. The date of diagnosis was defined as the date of the first contact. Danish versions of ICD-10 codes associated with the contacts determined the disease subtype (CD ICD-10 code: DK50; UC ICD-10 code: DK51). If both CD and UC were recorded, the disease subtype was classified as CD[41].

### Covariates
To describe the study population, we retrieved information on maternal, paternal, and child characteristics from DNBC telephone interviews and the National Patient Registry. The following variables were assessed in the telephone interviews: maternal pre-pregnancy body mass index (kg/m$^2$), maternal alcohol intake in second trimester (yes/no), maternal smoking in second trimester (yes/no), maternal nutritional supplement use in second trimester (yes/no), child's exclusive breastfeeding duration and any

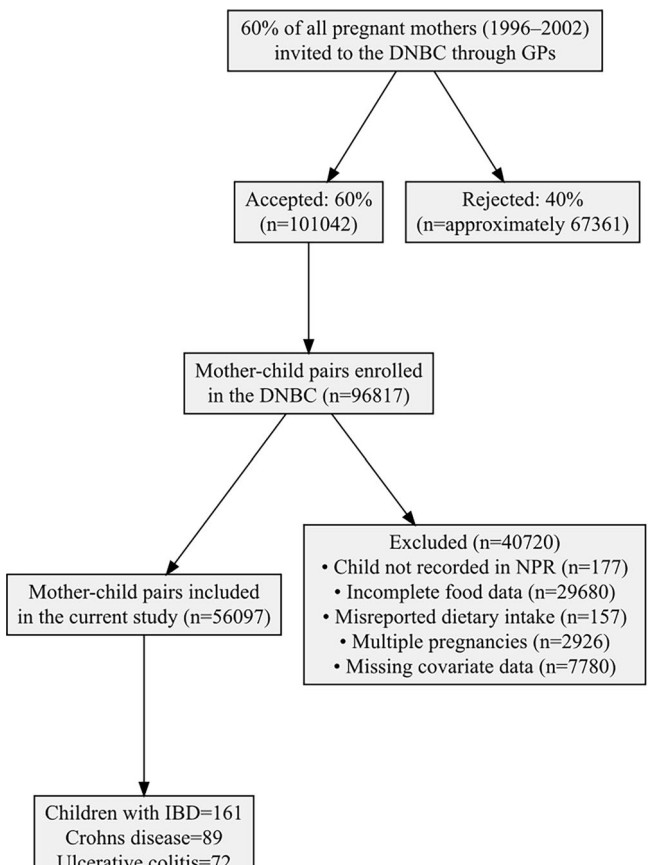

**Fig. 1 | Flow diagram of the study population selection.** 'Incomplete food data' refers to incomplete responses to the food frequency questionnaire. DNBC Danish National Birth Cohort, GP general practitioner, IBD inflammatory bowel disease, NPR National Patient Registry.

breastfeeding duration (days, truncated at 180 days), and maternal educational level (primary/secondary or post-secondary/tertiary). Maternal educational level follows definitions by the International Standard Classification of Education[42], where levels 0–2 reflect primary education, levels 3–4 reflect secondary and post-secondary education, and levels 5–8 reflect tertiary education. The following variables were retrieved from the National Patient Registry: maternal age at childbirth (years), maternal or paternal IBD diagnosis (yes/no), maternal antibiotics use in pregnancy (0 courses/ 1–2 courses/≥3 courses), child's antibiotics use in the first year of life (yes/ no), child's sex (boy/girl), delivery mode (vaginal/cesarean), and preterm delivery (yes/no). In addition, we also evaluated maternal diet quality during pregnancy using the Healthy Eating Index, which was previously developed within the DNBC based on maternal adherence to Danish official dietary recommendations[43].

**Statistical analyses**

R was used for all statistical analyses with a significance level of $p < 0.05$. Characteristics of mothers and their children were described according to clusters of maternal dietary patterns. Data with fewer than 10 observations were presented as <10 to comply with General Data Protection Regulation guidelines and protect individual privacy. Continuous variables were expressed as medians and interquartile range (IQR), and categorical variables were expressed as counts with corresponding percentages. Depending on the data type, Kruskal-Wallis or $X^2$ tests compared the maternal and offspring characteristics across the clusters of maternal dietary patterns.

Pediatric-onset IBD was ascertained from the date of birth to the date of diagnosis, death, emigration, or 18th birthday, whichever occurred first. Cox proportional hazards model estimated the offspring's risk of

pediatric-onset IBD associated with maternal dietary patterns, using the most common pattern as reference. Furthermore, associations were explored separately for the two IBD subtypes, CD and UC. The proportional hazards assumption was tested for violation using the Schoenfeld residuals test, and none of the analyses violated this assumption (all p>0.05).

Results were presented as hazard ratios with corresponding 95% confidence intervals before and after adjusting for covariates. The adjusted analyses included the following covariates, which were selected a priori based on knowledge about diet-related risk factors for IBD: maternal educational level, pre-pregnancy body mass index, smoking during pregnancy, nutritional supplement use during pregnancy, antibiotic use during pregnancy, energy intake during pregnancy, and parental IBD diagnosis. Another model further adjusted for offspring's antibiotics use during the first life year. Those with missing covariate data were excluded, and their proportion was reported in the results section (Fig. 1).

**Ethical approval**

The DNBC was approved by the National Committee on Health Research Ethics in Denmark (ref. no. [KF] 01-471/94) and conducted according to the guidelines of the Declaration of Helsinki. All participants gave written informed consent at enrollment on behalf of themselves and their unborn child, thereby giving permission for researchers to use their information in medical records and registers for long-term follow-up. The register data are protected by the Danish Act on Processing of Personal Data and are accessed through application to and approval from the Danish Data Protection Agency and the Danish Health Data Authority. The Center for Molecular Prediction of Inflammatory Bowel Disease (PREDICT) has got ethical approval to work on the data by The Danish Council on Ethics (H-20048987). No new ethical approval was needed to conduct the current study, but authorization to use the DNBC data was sought and approved by the DNBC Reference Group.

**Patient and public involvement**

This study was based on historical data. However, the Danish Colitis-Crohn Association (CCF) was consulted regarding the relevance of the study. They found it to be highly relevant and have expressed their support for the project.

## Results
### Mother–child pairs

The DNBC enrolled 96,817 liveborn mother-child pairs, of which 40,720 were excluded from our analyses for reasons detailed in Fig. 1. Baseline characteristics of the included and excluded populations are presented in Supplementary Table 2. Accordingly, the final study population consisted of 56,097 mother–child pairs. By the age of 18 years, 161 children (0.29%) had developed pediatric-onset IBD, with a median (IQR) age at diagnosis of 15.0 (13.0–17.0) years. Among these, 89 children (0.16%) were diagnosed with pediatric-onset CD, and 72 (0.13%) with pediatric-onset UC. Kaplan-Meier curves illustrating cumulative incidence rates are presented in Supplementary Fig. 1.

Table 1 outlines the study population's characteristics. At childbirth, the mothers had a median (IQR) age of 30.0 (27.0–33.0) years. Their median (IQR) diet quality score was 22.2 (17.8–27.1), and their median (IQR) daily energy intake was 9831 (8277–11,616) kJ/day. Regarding educational attainment, 48.6% had tertiary education. During pregnancy, 50.7% drank alcohol, 15.9% smoked, 97.9% used nutritional supplements, and 29.9% used antibiotics at least once. Among the children, 3.6% were born preterm, 41.4% received at least one course of antibiotics during the first year of life, and 1.2% had a parent diagnosed with IBD.

### Maternal dietary patterns during pregnancy

K-means cluster analysis identified five different clusters of maternal dietary patterns during pregnancy (Fig. 2, Supplementary Table 3). Cluster 1 (n = 8388) had a diverse dietary pattern with high intakes of most food groups, including meat, fish, vegetables, legumes, fruits, potatoes, and

**Table 1 | Characteristics of the study population according to maternal dietary patterns during pregnancy[a]**

| | Total (n = 56,097) | Diverse dietary pattern (n = 8388) | Non-recommended dietary pattern (n = 14,110) | Mediterranean dietary pattern (n = 14,700) | Animal-based dietary pattern (n = 3808) | Western dietary pattern (n = 15,091) |
|---|---|---|---|---|---|---|
| **Maternal age at birth, years** | | | | | | |
| Median (IQR) | 30.0 (27.0–33.0) | 31.0 (28.0–34.0) | 29.0 (26.0–32.0) | 31.0 (28.0–34.0) | 30.0 (27.0–33.0) | 29.0 (26.0–32.0) |
| **Maternal educational level** | | | | | | |
| Primary, n (%) | 6820 (12.2%) | 844 (10.1%) | 2253 (16.0%) | 1067 (7.3%) | 497 (13.1%) | 2159 (14.3%) |
| Secondary or post-secondary, n (%) | 22021 (39.3%) | 3034 (36.2%) | 6339 (44.9%) | 4224 (28.7%) | 1492 (39.2%) | 6932 (45.9%) |
| Tertiary, n (%) | 27256 (48.6%) | 4510 (53.8%) | 5518 (39.1%) | 9409 (64.0%) | 1819 (47.8%) | 6000 (39.8%) |
| **Maternal pre-pregnancy BMI, kg/m²** | | | | | | |
| Median (IQR) | 22.65 (20.69–25.35) | 22.41 (20.57–24.91) | 23.18 (21.01–26.40) | 22.05 (20.42–24.30) | 22.31 (20.48–24.80) | 23.03 (20.83–25.91) |
| **Maternal alcohol intake in 2ⁿᵈ trimester** | | | | | | |
| Yes, n (%) | 28424 (50.7%) | 5408 (64.5%) | 5129 (36.4%) | 8652 (58.9%) | 1911 (50.2%) | 7324 (48.5%) |
| No, n (%) | 27673 (49.3%) | 2980 (35.5%) | 8981 (63.6%) | 6048 (41.1%) | 1897 (49.8%) | 7767 (51.5%) |
| **Maternal smoking in 2ⁿᵈ trimester** | | | | | | |
| Yes, n (%) | 8915 (15.9%) | 1302 (15.5%) | 2559 (18.1%) | 1247 (8.5%) | 734 (19.3%) | 3073 (20.4%) |
| No, n (%) | 47182 (84.1%) | 7086 (84.5%) | 11551 (81.9%) | 13453 (91.5%) | 3074 (80.7%) | 12018 (79.6%) |
| **Maternal diet quality in 2ⁿᵈ trimester** | | | | | | |
| Median (IQR) | 22.2 (17.8–27.1) | 24.8 (21.2–28.4) | 20.4 (16.1–25.6) | 26.3 (22.1–31.1) | 22.0 (17.7–27.0) | 18.7 (15.4–22.3) |
| **Maternal nutritional supplement use in 2ⁿᵈ trimester** | | | | | | |
| Yes, n (%) | 54919 (97.9%) | 8223 (98.0%) | 13759 (97.5%) | 14482 (98.5%) | 3720 (97.7%) | 14735 (97.6%) |
| No, n (%) | 1178 (2.1%) | 165 (2.0%) | 351 (2.5%) | 218 (1.5%) | 88 (2.3%) | 356 (2.4%) |
| **Maternal antibiotics use in pregnancy** | | | | | | |
| 0 courses, n (%) | 39316 (70.1%) | 5827 (69.5%) | 9728 (68.9%) | 10791 (73.4%) | 2626 (69.0%) | 10344 (68.5%) |
| 12 courses, n (%) | 13092 (23.3%) | 1976 (23.6%) | 3416 (24.2%) | 2967 (20.2%) | 931 (24.4%) | 3802 (25.2%) |
| ≥3 courses, n (%) | 3689 (6.6%) | 585 (7.0%) | 966 (6.8%) | 942 (6.4%) | 251 (6.6%) | 945 (6.3%) |
| **Child's sex** | | | | | | |
| Girl, n (%) | 27284 (48.6%) | 4166 (49.7%) | 6809 (48.3%) | 7158 (48.7%) | 1811 (47.6%) | 7340 (48.6%) |
| Boy, n (%) | 28813 (51.4%) | 4222 (50.3%) | 7301 (51.7%) | 7542 (51.3%) | 1997 (52.4%) | 7751 (51.4%) |
| **Parental IBD diagnosis** | | | | | | |
| Yes, n (%) | 653 (1.2%) | 103 (1.2%) | 182 (1.3%) | 176 (1.2%) | 34 (0.9%) | 158 (1.0%) |
| No, n (%) | 55444 (98.8%) | 8285 (98.8%) | 13928 (98.7%) | 14524 (98.8%) | 3774 (99.1%) | 14933 (99.0%) |
| **Preterm delivery** | | | | | | |
| Yes (<37 weeks), n (%) | 2026 (3.6%) | 304 (3.6%) | 531 (3.8%) | 499 (3.4%) | 105 (2.8%) | 587 (3.9%) |
| No (≥37 weeks), n (%) | 53355 (95.1%) | 7963 (94.9%) | 13458 (95.4%) | 13934 (94.8%) | 3628 (95.3%) | 14372 (95.2%) |
| Data missing, n (%) | 716 (1.3%) | 121 (1.4%) | 121 (0.9%) | 267 (1.8%) | 75 (2.0%) | 132 (0.9%) |
| **Child's antibiotics use in first year of life** | | | | | | |
| Yes, n (%) | 23208 (41.4%) | 3363 (40.1%) | 6142 (43.5%) | 5320 (36.2%) | 1598 (42.0%) | 6785 (45.0%) |
| No, n (%) | 32889 (58.6%) | 5025 (59.9%) | 7968 (56.5%) | 9380 (63.8%) | 2210 (58.0%) | 8306 (55.0%) |
| **Any breastfeeding duration,[b] days** | | | | | | |
| Median (IQR) | 180 (134–180) | 180 (164–180) | 180 (120–180) | 180 (180–180) | 180 (150–180) | 180 (120–180) |
| Data missing, n (%) | 10634 (19.0%) | 1467 (17.5%) | 2368 (16.8%) | 2540 (17.3%) | 1886 (49.5%) | 2373 (15.7%) |
| **Exclusive breastfeeding duration,[b] days** | | | | | | |
| Median (IQR) | 120 (90–150) | 120 (104–150) | 120 (74–132) | 120 (120–150) | 0 (0–7) | 120 (87–134) |
| Data missing, n (%) | 21372 (38.1%) | 2724 (32.5%) | 4830 (34.2%) | 4947 (33.7%) | 3803 (99.9%) | 5068 (33.6%) |
| **Mode of delivery** | | | | | | |
| Vaginal, n (%) | 47827 (85.3%) | 7132 (85.0%) | 11902 (84.4%) | 12532 (85.3%) | 3344 (87.8%) | 12917 (85.6%) |
| Cesarean, n (%) | 8252 (14.7%) | 1252 (14.9%) | 2204 (15.6%) | 2164 (14.7%) | 462 (12.1%) | 2170 (14.4%) |

BMI body mass index, IBD inflammatory bowel disease.
[a]Data are expressed as median (Q1–Q3) for continuous variables and counts (%) for categorical variables.
[b]Truncated at 180 days.

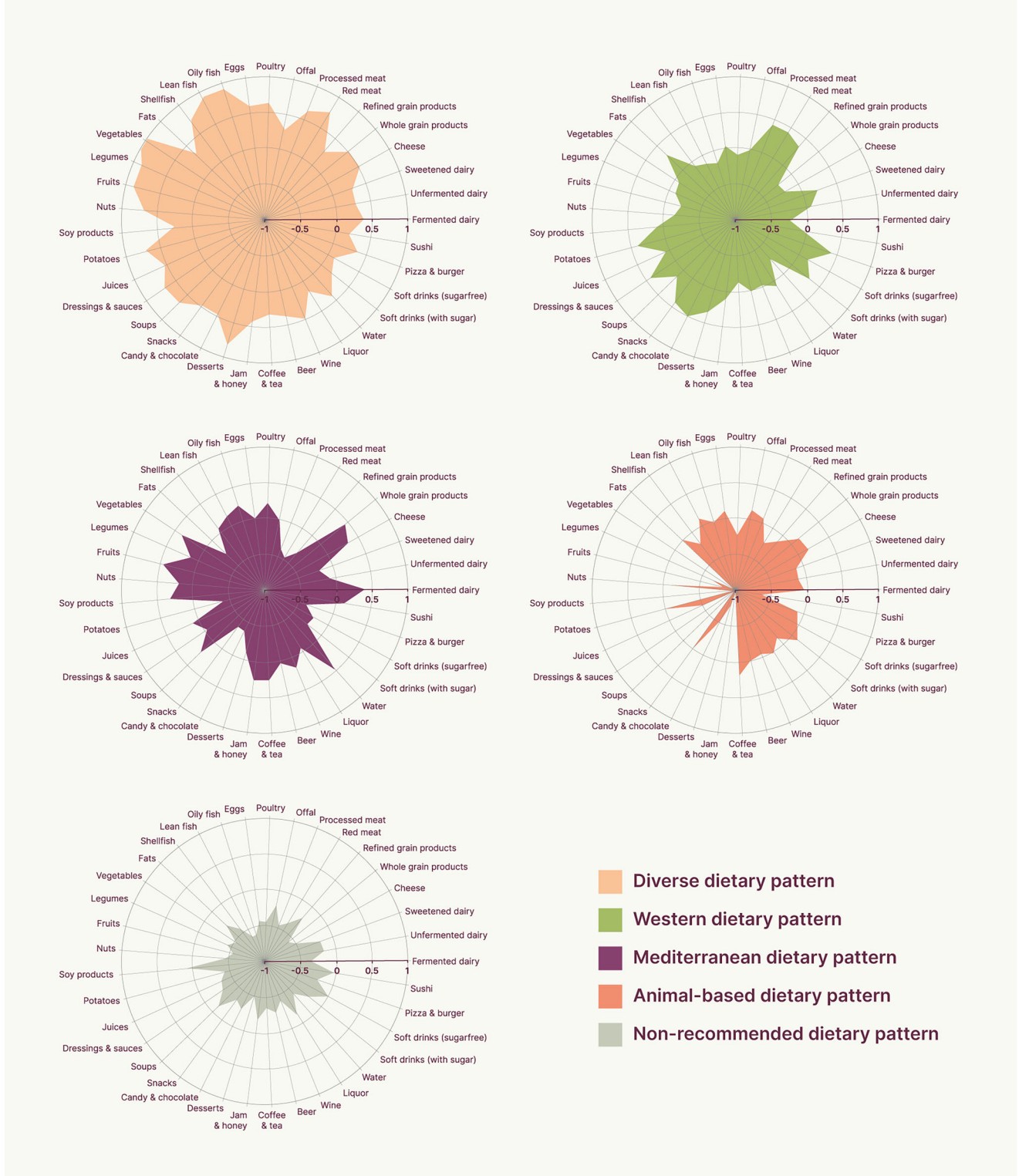

**Fig. 2 | Spider plot of the five maternal dietary patterns identified with k-means cluster analysis (Diverse = 8388; Non-recommended = 14,110; Mediterranean = 14,700; Animal-based = 3808; Western = 15,091).** The values represent z-score normalized daily intakes of 37 food groups across the five clusters.

desserts. Cluster 2 (n = 14,110) had a non-recommended dietary pattern with below-average intakes of most food groups. Cluster 3 (n = 14,700) was characterized as following a Mediterranean dietary pattern due to their above-average intakes of whole grain products, cheese, fermented dairy, fruits, vegetables, and fish. Cluster 4 (n = 3808) had an animal-based dietary pattern composed of most animal-based foods, although they also consumed other food groups, such as whole grain products and soft drinks with

sugar. Cluster 5 (n = 15,091) was characterized as following a Western dietary pattern with high intakes of processed meat, refined grain products, snacks, candies, chocolate, potatoes, pizzas, burgers, and sugar-free soft drinks. This dietary pattern was also the most common and was therefore used as the reference in further analyses.

Compared to mothers with a Western dietary pattern (Table 1), those following diverse, Mediterranean, or animal-based dietary patterns were

**Table 2 | Macronutrient intake according to maternal dietary patterns during pregnancy[a]**

| | Total (n = 56,097) | Diverse dietary pattern (n = 8388) | Non-recommended dietary pattern (n = 14,110) | Mediterranean dietary pattern (n = 14,700) | Animal-based dietary pattern (n = 3808) | Western dietary pattern (n = 15,091) |
|---|---|---|---|---|---|---|
| Energy (kJ/day) | 9831 (8277–11616) | 11924 (10414–13724) | 8274 (7009–9727) | 9498 (8253–10914) | 9427 (7974–11008) | 10670 (9244–12360) |
| Protein (g/day) | 90.3 (75.7–106.6) | 111.0 (97.0–127.9) | 75.8 (63.4–89.7) | 90.6 (78.3–104.1) | 86.2 (72.5–101.2) | 93.7 (81.0–108.4) |
| Fat (g/day) | 78.9 (62.2–100.6) | 97.9 (81.7–120.7) | 65.3 (51.7–83.1) | 70.3 (58.1–86.1) | 72.7 (57.7–91.1) | 92.0 (75.6–113.7) |
| Saturated fat (g/day) | 31.8 (24.1–42.1) | 38.8 (31.0–49.4) | 27.0 (20.5–36.0) | 27.2 (21.5–35.0) | 28.5 (21.7–37.5) | 38.2 (30.2–48.9) |
| Monounsaturated fat (g/day) | 24.7 (19.4–31.8) | 31.2 (25.9–38.5) | 20.3 (16.1–25.9) | 21.9 (18.0–27.0) | 22.3 (17.5–28.3) | 29.2 (23.9–36.0) |
| Polyunsaturated fat (g/day) | 11.8 (9.5–14.6) | 14.9 (12.7–17.8) | 9.3 (7.5–11.5) | 11.4 (9.5–13.5) | 11.9 (9.5–14.7) | 12.9 (10.7–15.5) |
| Omega-3 polyunsaturated fat (g/day) | 0.6 (0.5–0.8) | 0.9 (0.7–1.1) | 0.5 (0.4–0.6) | 0.7 (0.5–0.8) | 0.6 (0.5–0.8) | 0.7 (0.5–0.8) |
| Omega-6 polyunsaturated fat (g/day) | 2.5 (1.9–3.2) | 3.1 (2.5–3.8) | 2.1 (1.6–2.7) | 2.2 (1.8–2.9) | 2.3 (1.7–2.9) | 2.9 (2.3–3.5) |
| Carbohydrate (g/day) | 323.9 (269.6–384.3) | 384.6 (330.5–447.3) | 274.4 (228.8–325.8) | 322.5 (274.1–375.6) | 317.4 (267.4–372.6) | 341.1 (290.9–397.8) |
| Added sugar (g/day) | 40.7 (27.3–61.7) | 49.2 (34.9–70.4) | 33.9 (22.5–52.1) | 34.4 (23.8–48.9) | 37.4 (24.1–61.7) | 51.4 (35.9–76.1) |
| Dietary fiber (g/day) | 26.6 (20.7–33.1) | 31 (26.1–38.1) | 21.6 (16.3–27.7) | 28.9 (23.3–35.3) | 26.1 (20.7–32.4) | 25.9 (20.7–32.0) |

[a]Data are expressed as median (Q1–Q3) according to maternal dietary patterns.

older at childbirth, had a higher educational level, higher diet quality, and lower body mass index. Mothers with a Mediterranean dietary pattern also smoked less (8.5% versus 20.4%) and their children had a lower antibiotic use during early life (36.2% versus 45.0%) compared to mothers with a Western dietary pattern. Mothers with a diverse dietary pattern had the highest intake of all macronutrients, except added sugar, compared to mothers with other dietary patterns. Conversely, mothers with a non-recommended dietary pattern had the lowest intake of energy and all macronutrients compared to mothers with other dietary patterns (Table 2).

**Maternal dietary patterns and offspring's risk of pediatric-onset inflammatory bowel disease**

A maternal diverse dietary pattern during pregnancy was associated with a 45% lower risk of pediatric-onset IBD in offspring compared to a Western dietary pattern (adjusted hazard ratio: 0.55; 95% confidence interval: 0.31–0.97). Analyzing the two IBD subtypes separately showed a non-significant lower risk of both pediatric-onset CD (adjusted hazard ratio: 0.67; 95% confidence interval: 0.31–1.45) and pediatric-onset UC (adjusted hazard ratio: 0.44; 95% confidence interval: 0.19–1.03). Furthermore, compared to a Western dietary pattern, the other three dietary patterns showed no significant association with the risk of pediatric-onset IBD in offspring, including separate analyses for CD and UC (Table 3). Additionally adjusting for offspring's antibiotics use during the first year of life had no impact on the results (Supplementary Table 4).

## Discussion

In this extensive nationwide prospective cohort study of 56,097 mother-child pairs followed over 18 years, we demonstrated that a maternal diverse dietary pattern, compared to a Western dietary pattern, during pregnancy was associated with a 45% lower risk of pediatric-onset IBD in the offspring. This association was found for both pediatric-onset CD and UC when analyzing these disease subtypes separately, but neither were statistically significant. Other dietary patterns (Mediterranean, animal-based, and non-recommended) during pregnancy did not show significant associations with offspring's risk of pediatric-onset IBD, CD, or UC compared to a Western dietary pattern.

Our findings support previous results from animal studies suggesting that maternal diet during pregnancy may influence offspring's IBD susceptibility through effects on intestinal development, microbiota, permeability, and inflammatory response[17–25]. A recent birth cohort study from Norway similarly reported that higher diet diversity scores during pregnancy were associated with a reduced UC risk in offspring[26]. However, the Norwegian study quantified maternal dietary patterns a priori based on the variety in intakes of grains, fruits, vegetables, and animal-based products[26], whereas our empirically derived maternal dietary patterns are not necessarily indicative of diet diversity, although we also see a lower risk in the group characterized by high intakes across most food groups. Thus, while not directly comparable, both studies underscore a potential protective effect of consuming a variety of food groups during pregnancy.

Adherence to a diverse dietary pattern is the most widely recognized dietary recommendation for achieving overall health[44]. Incorporating various food groups into the diet ensures adequate intake of essential nutrients[45], which is particularly crucial during pregnancy due to their important roles in offspring's early development and long-term health[46,47]. In contrast, animal models suggest that a Western-style diet may persistently program the fetal immune system toward a proinflammatory state[48]. Hence, the protective association observed for the diverse dietary pattern, compared to the Western dietary pattern, may reflect a mitigation of these adverse immune effects.

Moreover, a diverse dietary pattern promotes gut microbiota diversity[49], which could contribute to IBD prevention, as individuals with IBD commonly exhibit reduced microbial diversity[50,51]. This influence may extend to the offspring, as the maternal microbiota shapes the infant's gut microbiota establishment[52–54]. A potential mechanism is through the vertical transfer of maternal microbes at delivery, with the delivery method being a

**Table 3 | Maternal dietary patterns during pregnancy and risk of pediatric-onset inflammatory bowel disease in the offspring[a]**

| | Diverse dietary pattern (n = 8388) | Non-recommended dietary pattern (n = 14,110) | Mediterranean dietary pattern (n = 14,700) | Animal-based dietary pattern (n = 3808) | Western dietary pattern (n = 15,091) |
|---|---|---|---|---|---|
| **Inflammatory bowel disease** | | | | | |
| Cases | 16 | 46 | 39 | 9 | 51 |
| Unadjusted HR (95% CI) | 0.58 (0.33–1.01) | 0.97 (0.65–1.45) | 0.81 (0.53–1.23) | 0.72 (0.35–1.45) | Reference |
| Adjusted HR (95% CI)[b] | 0.55 (0.31–0.97) | 1.08 (0.70–1.66) | 0.87 (0.56–1.35) | 0.76 (0.37–1.56) | Reference |
| **Crohn's disease** | | | | | |
| Cases | <10[c] | 31 | 23 | <10[c] | 24 |
| Unadjusted HR (95% CI) | 0.69 (0.32–1.48) | 1.40 (0.82–2.38) | 1.02 (0.57–1.80) | 0.34 (0.08–1.43) | Reference |
| Adjusted HR (95% CI)[b] | 0.67 (0.31–1.45) | 1.41 (0.79–2.53) | 0.98 (0.54–1.78) | 0.34 (0.08–1.44) | Reference |
| **Ulcerative colitis** | | | | | |
| Cases | <10[c] | 15 | 16 | <10[c] | 27 |
| Unadjusted HR (95% CI) | 0.48 (0.21–1.10) | 0.60 (0.32–1.13) | 0.63 (0.34–1.17) | 1.05 (0.46–2.42) | Reference |
| Adjusted HR (95% CI)[b] | 0.44 (0.19–1.03) | 0.73 (0.37–1.44) | 0.77 (0.40–1.48) | 1.20 (0.52–2.80) | Reference |

*CI* confidence interval, *HR* hazard ratio.

[a]HRs refer to offspring's disease risk associated with the specific dietary pattern relative to the Western dietary pattern.

[b]Adjusted for maternal educational level, pre-pregnancy body mass index, smoking during pregnancy, nutritional supplement use during pregnancy, energy intake during pregnancy, antibiotics use during pregnancy, and parental inflammatory bowel disease diagnosis.

[c]Data presented as <10 to comply with General Data Protection Regulation guidelines and protect individual privacy.

key factor[55]. Indeed, a birth cohort study from the United States found that associations between maternal diet and infant gut microbiota differed for vaginally versus cesarean-delivered infants, indicating that delivery method could modify the microbiota-mediated effects of maternal diet on offspring IBD risk[56]. Nonetheless, delivery method was not adjusted for in our analyses because there is no evidence that it is directly linked to maternal dietary patterns during pregnancy, making confounding unlikely.

The observed association between a diverse maternal dietary pattern during pregnancy and offspring's IBD risk may not be solely explained by in-utero exposures. The offspring's own dietary pattern during childhood also likely plays a role, potentially formed by maternal dietary patterns during pregnancy. Another study from the DNBC found that maternal diet quality during pregnancy predicts offspring's diet quality at age 14[43], and more broadly, research shows that children's dietary habits often mirror those of their parents[57,58]. Therefore, the dietary patterns identified among the mothers may have been adopted by their children, thereby indirectly influencing IBD risk through postnatal exposures. This also aligns with prior studies in children showing an association between various elements of childhood diet and lower IBD risk, including high intakes of fruits, vegetables, and fish, and low intakes of sugar-sweetened beverages and fatty foods[59]. Likewise, breastfeeding and infant feeding practices could also mediate part of the observed association, mainly if maternal dietary patterns persisted postpartum. Maternal dietary patterns influence the maternal breast milk composition[60,61], which in turn might shape offspring's immune development and gut microbiota colonization via breastfeeding[62,63]. However, this is less expected to explain our findings, as previous results from the DNBC showed no association between breastfeeding and offspring's IBD risk[64].

Our study has several strengths. First, its prospective design, large sample size, and long follow-up period collectively capture the longitudinal impact of maternal dietary patterns on offspring's disease development. Second, clustering maternal food intake into dietary patterns mitigates type I error from multiple comparisons that would have arisen if all 37 food groups were analyzed independently. Third, unlike predefined indices or scores, empirically derived dietary patterns are less affected by a priori assumptions

about diet and health[29]. Although principal component analysis is more commonly employed to derive such patterns[65], we applied k-means cluster analysis, which has the advantage of identifying mutually exclusive dietary patterns for comparison regarding offspring's disease risk. Notably, previous studies show that dietary patterns derived from cluster and principal component analyses are largely comparable[66,67], suggesting that both methods capture substantial variation in dietary behaviors. Fourth, dietary pattern analysis provides a more holistic perspective on maternal diet by considering the complex interrelationships among food groups. Compared to single food or nutrient-based analyses, dietary patterns are more strongly associated with gut microbiota variations[68], a potential key player in IBD pathogenesis[69–71] that is thought to mediate the relationship between diet and IBD[72]. Lastly, dietary pattern-based recommendations may seem easier to implement than nutrient-specific advice, which often requires a deeper understanding of food composition.

This study also has some potential limitations. Residual confounding cannot be ruled out due to its observational design, despite adjusting for multiple relevant covariates. The small number IBD cases may have limited statistical power and could partly explain the non-significant associations observed in subtype-specific analyses of CD and UC. Selection bias may also exist from the low participation rates, although a previous study from the DNBC suggested that non-participation had minimal impact on different exposure-outcome associations[73]. Furthermore, dietary pattern analysis involves multiple subjective decisions, such as choosing the number of food groups for cluster analysis and naming the identified patterns. To minimize observer bias, we used standard food group classifications[34] for objectivity and transparency. Because the dietary patterns are based on z-scores unadjusted for total energy intake, they may reflect absolute intake levels; however, we adjusted for energy intake in the Cox regression models to account for this. Another limitation is that maternal diet was assessed only in the second trimester, although its impact on fetal development may vary across trimesters[74]. Still, this is unlikely to affect the study's validity, as dietary patterns remain relatively stable over time[75] and show minimal variation from pre-conception through pregnancy despite some changes in intakes of specific food groups[76–78]. Since our study focused on dietary

patterns, the additional information gained from repeated assessments may therefore not outweigh the burden to participants and investigators. Finally, dietary pattern analysis remains complementary to analyses of single foods and nutrients, which are key to understanding the biological mechanisms behind the observed associations[28]. Thus, our findings might reflect differences in intakes of specific foods or nutrients among mothers adhering to a diverse compared to a Western dietary pattern, but further investigation is needed to clarify this.

## Conclusion

This nationwide prospective cohort study demonstrates that a maternal diverse dietary pattern, compared to a Western dietary pattern, during pregnancy is associated with a reduced risk of pediatric-onset IBD in the offspring. In contrast, a Mediterranean, animal-based, or non-recommended dietary pattern during pregnancy showed no association with offspring's disease risk compared to a Western dietary pattern.

These findings offer insights into the role of maternal a posteriori dietary patterns in the etiology of pediatric-onset IBD, underscoring their significance for future generations' health. Given the limited research in the area, however, further studies are needed to identify specific diets that may influence IBD development in offspring. This includes studying maternal diet from other perspectives to clarify whether the observed associations are driven by elements unique to each of the five identified dietary patterns.

## Data availability

The datasets analyzed during the current study are not publicly available due to privacy protection of participants in the DNBC. Researchers who wish to access the data can apply through the DNBC Secretariat. Detailed information on the application process, including conditions of use and data access agreements, is available at the DNBC webpage (https://www.dnbc.dk/access-to-dnbc-data). The numerical results underlying Fig. 2 are presented in Supplementary Table 3.

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

## Acknowledgements

The work was financially supported by the Danish National Research Foundation to TJ (DNRF148) and The European Crohn's and Colitis Organisation as part of the D-ECCO research grant to MBF (D-grant/ECCO). The DNBC was established with a significant grant from the Danish National Research Foundation. Additional support was obtained from the Danish Regional Committees, the Pharmacy Foundation, the Egmont Foundation, and other minor grants. March of Dimes Birth Defects Foundation (6-FY-96-0240, 6-FY97-0553, 6-FY97-0521, 6-FY00-407), the European Union (QLK1-2000-00083), the Danish Medical Research Foundation (9601842 and 22-03-0536), the Health Foundation (11/263-96), and the Heart Foundation (96-2-4-83-22450) supported the collection, development, and elaboration of the dietary data. The study funders had no role in study design, data collection, data analysis, data interpretation, writing of the report, or decision to submit the paper for publication.

## Author contributions

OMA, MBF, MJ, and TJ were responsible for conceptualization, resources, and funding acquisition. SFO, AAB, and TIH were responsible for data curation. OMA was responsible for planning, methodology, and visualization. OMA and AVH were responsible for formal analysis. MBF and TJ were responsible for project administration and supervision. OMA was responsible for writing the original draft, and all authors contributed equally to revising the final draft. All authors had full access to the data and had final responsibility for the decision to submit for publication. The corresponding author attests that all listed authors meet authorship criteria and that no others meeting the criteria have been omitted.

## Competing interests

MJ has received consulting/advisory board fees from Ferring, Takeda, AbbVie, PharmaCosmos and Tillots Pharma; speaker's fees from Tillotts Pharma, MSD, Ferring and Takeda. TJ has received consultancy fees from Ferring and Pfizer. MJ has received research grants for other investigator-driven studies from Takeda, and the Novo Nordisk Foundation (NNF23Oc0081717). All other authors declare no competing interests. The authors declare that there are no other financial or non-financial relationships or activities that could appear to have influenced the submitted work.
