## [Transparent Peer Review file · Communications Medicine]

Maternal mid-pregnancy dietary patterns and inflammatory bowel disease in offspring from a prospective cohort study

Corresponding Author: Ms Olivia Mariella Anneberg

Version 0:

Reviewer comments:

Reviewer #1

(Remarks to the Author)

Why was the manuscript organized that way?! Introduction followed by the results, followed by the discussion and then methods!! Never seen anything like this before which made the review process difficult! I looked at authors guidelines and they recommended reporting using STROBE statement.

The authors claimed that this is a prospective study. I am not sure that this is true. I see that the data were collected prospectively as a part of the Danish National birth cohort study but was looking at PIBD in the offspring a clear objective in the original proposal?

The diet in the study is a second trimester diet. The authors claimed that second trimester diet is likely to be all through pregnancy. They cited a ref. to support their claim that I failed to find what supported their claim within that reference. I would strongly recommend adding "second trimester diet" to the title, abstract and methodology with keeping this part as a limitation within the discussion.

The authors used dietary cluster analysis. I wonder how they avoided drawbacks associated with this analysis especially the uncertainty of individual classification and subjective selection of the diet cluster. What clustering algorithms did they use to determine the similarity of individuals? How did they come with the definition of these the 5 clusters? Source/ref of this classification? Any validation of these clusters? I feel there is a great chance of cluster overlap. How did they avoid any possibility of overlap?

How did they validate the algorithm for the diagnosis of PIBD?

What package of R software for analysis did they use?

Any chance of additional analysis using combined clusters; mainly diverse plus Mediterranean vs the rest?
I am not sure that I understand the numbers in supp table 2?

Any chance of adding social class and family income to the model of Cox analysis?

A minor issue is the statement in the introduction "Approximately 25% of IBD cases have a pediatric onset". The ref. they cite took that statement from a review article. I think this is a myth that has been cited many times without a validated source/ref. based on a good research/population-based study. It originally came from a textbook from 1980s! Please revise.

Some references were not in the journal format (e.g. Ref 27 lacking the year, ref 28)

Reviewer #2

(Remarks to the Author)

Brief summary of the manuscript:

The investigators examined associations between maternal dietary patterns during pregnancy and offspring's pediatric-onset IBD risk among 56,097 mother-child pairs from the Danish National Birth Cohort. They derived dietary patterns using k-means cluster analyses on maternal z-score normalized intakes of 37 food groups, assessed with a food frequency questionnaire at gestational week 25. They identified 5 dietary patterns - diverse, non-recommended, Mediterranean, animal-based, and Western. They found that a offspring of mothers who consumed a diverse dietary pattern had a 45% lower risk of pediatric-onset IBD compared to those of mothers who consumed a Western dietary pattern.

Overall impression of the work:

This was a significant endeavour to collect 56,097 mother-child pairs of food frequency data and to have that number of offspring followed to identify pediatric-onset IBD cases. Their method to derive dietary patterns using an a posteriori approach rather than an a priori approach brings a unique angle to this type of research.

A few points if the authors could elaborate or address, would strengthen the manuscript

- 1) is the one time food frequency questionnaire sufficient to reflect the dietary intake throughout pregnancy and into breastfeeding?
- 2) delivery method - although possibly a controversial risk factor - there was no mention of the variable of delivery method being analyzed
- 3) they mention "breastfeeding duration" was assessed, but did not mention about formula feeding (which formula feeding or non-exclusive breastfeeding could affect offspring risk of developing IBD)

Reviewer #3

(Remarks to the Author)

The authors are to be commended on this well designed and well written study that I thoroughly enjoyed reading. Only one question: It does not appear that there was adjustment for child's antibiotics use in the first year of life, which is known to be associated with pediatric IBD. Did the authors look at this?

Minor:

- Typo in Figure 2 spider plots for sugar free soft drinks (says "sugerfree")
- The link in the data availability statement does not work (says "Page not found")

Version 1:

Reviewer comments:

Reviewer #2

(Remarks to the Author)

Thank you for addressing the reviewer comments. The revised manuscript reads nicely

Reviewer #3

(Remarks to the Author)

No additional comments from me.

Dear Reviewers,

We thank you for your thorough evaluation of our manuscript and for the constructive comments. We have taken all the comments into account and updated the manuscript accordingly. Please see the marked changes in the manuscript, as well as our point-by-point responses below.

We sincerely appreciate the time and effort you have invested in reviewing our work and find that your feedback has significantly improved it.

Sincerely,

Olivia Mariella Anneberg, Sjurdur Frodi Olsen, Anne Vinkel Hansen, Mette Julsgaard, Anne Ahrendt Bjerregaard, Thorhallur Ingi Halldorsson, Tine Jess, & Maiara Brusco De Freitas.

AUTHOR RESPONSE LETTER

Reviewer #1

Why was the manuscript organized that way?! Introduction followed by the results, followed by the discussion and then methods!! Never seen anything like this before which made the review process difficult! I looked at authors guidelines and they recommended reporting using STROBE statement.

- **Response:** Thank you for all your valuable comments. The manuscript was originally formatted for submission to another Nature journal, which required the manuscript to be organized that way. We have now restructured the manuscript to follow the Communications Medicine author guidelines, with the Methods section placed between the Results and Discussion.

The authors claimed that this is a prospective study. I am not sure that this is true. I see that the data were collected prospectively as a part of the Danish National birth cohort study but was looking at PIBD in the offspring a clear objective in the original proposal?

- **Response:** It is correct that the data were prospectively collected as part of the Danish National Birth Cohort (DNBC). The cohort was established in the 1990s with the overall aim of examining early life exposures and their long-term impact on disease susceptibility. Although not specifically designed for inflammatory bowel disease (IBD), the idea behind the DNBC was to investigate all diseases with a possible origin in the early life period, including IBD and other immune-mediated diseases (Olsen J, et al. *Scand J Public Health* 2001;29:300–7). Therefore, our study fulfills the definition of a prospective cohort study, as the cohort was followed forward in time, with the exposure of interest (maternal diet during pregnancy) assessed before the outcome (offspring IBD).

The diet in the study is a second trimester diet. The authors claimed that second trimester diet is likely to be all through pregnancy. They cited a ref. to support their claim that I failed to find what supported their claim within that reference. I would strongly recommend adding “second trimester

diet” to the title, abstract and methodology with keeping this part as a limitation within the discussion.

- **Response:** This is a very important point since there might be different impacts of diet on offspring disease across the trimesters. Therefore, we agree that restricting dietary assessment to the second trimester is a limitation of our study, which we have now expanded upon in the Discussion (lines 343-348). Nonetheless, previous studies have shown that dietary patterns remain stable throughout pregnancy, despite some variation in specific food groups. For example, one of the cited studies concluded that pregnancy dietary patterns “do not vary significantly over time” and that “food intake information obtained at one point during pregnancy can provide reliable information of dietary pattern throughout pregnancy” (Cucó G, et al. *Eur J Clin Nutr* 2006;60(3):364–71). Based on this, we believe that assessing dietary patterns in the second trimester is sufficient to capture their potential impact on offspring IBD risk. To ensure clarity for the readers, we have specified in the title, abstract (line 25), and methods (line 96 + 117-118) that maternal diet was assessed in the second trimester.

The authors used dietary cluster analysis. I wonder how they avoided drawbacks associated with this analysis especially the uncertainty of individual classification and subjective selection of the diet cluster. What clustering algorithms did they use to determine the similarity of individuals? How did they come with the definition of these the 5 clusters? Source/ref of this classification? Any validation of these clusters? I feel there is a great chance of cluster overlap. How did they avoid any possibility of overlap?

- **Response:** Thank you for these thoughtful comments. To ensure clarity, we have elaborated more on the cluster analysis in the manuscript; please see lines 135-148. Regarding your questions about the methodology, we derived maternal dietary patterns using k-means cluster analysis, which assigns each mother-child pair to a single cluster, thus avoiding overlap. The optimal number of clusters (five) was determined using the Elbow method, a commonly used approach to identify the point where additional clusters will no longer improve the within-cluster homogeneity. The resulting clusters were defined entirely by the underlying dietary data. Our approach is consistent with established methodologies previously applied in studies using k-means cluster analysis to derive dietary patterns (Zhao J, et al. *Nutr J* 2021;20(1)). Internal validation was performed by repeating the algorithm with different initial seeds, producing stable cluster centroids.

How did they validate the algorithm for the diagnosis of PIBD?

- **Response:** IBD in offspring was identified in the Danish National Patient Register using Danish versions of ICD-10 codes (Crohn’s disease: DK50, ulcerative colitis: DK51) linked to hospital contacts (lines 151-158). Cases were defined as children with at least two IBD hospital contacts within a two-year period, including inpatient or combined inpatient/outpatient contacts. This definition is commonly used in register-based studies. We did not validate the diagnoses within this study population, but prior work from our group demonstrates high validity of the IBD diagnoses within the Danish patient register,

especially for patients with multiple registrations (Jacobsen HA, et al. *Clin Epidemiol* 2022;1099–1109).

What package of R software for analysis did they use?

- **Response:** We used the R-package ‘cluster’ to perform k-means cluster analysis (line 136).

Any chance of additional analysis using combined clusters; mainly diverse plus Mediterranean vs the rest?

- **Response:** We believe that the idea of combining the suggested clusters could be an interesting approach, particularly to increase statistical power. However, our aim was to identify naturally occurring, data-driven maternal dietary patterns and compare them regarding offspring’s IBD risk. Combining clusters post hoc would compromise this principle, as the resulting clusters would no longer reflect empirically derived dietary patterns.

I am not sure that I understand the numbers in supp table 2?

- **Response:** Thank you for addressing this important detail. The numbers in Supplementary Table 2 (Supplementary Table 3 in the revised version) represent the K-means cluster centroids for the five dietary clusters across the 37 analyzed food groups. These values indicate the mean intake of each food group (z-score normalized) within each cluster. To ensure clarity, we have updated the table heading and description to include this information. We have also explained this in the manuscript now, please see lines 143-144 and 222.

Any chance of adding social class and family income to the model of Cox analysis?

- **Response:** We agree that adding social class and family income is relevant, as these factors potentially could be important confounders. In our analysis, we already adjusted for maternal educational level, determined by classifying maternal occupation using ISCED-20211, as a proxy for socioeconomic status. Unfortunately, we did not have information on family income. However, income is generally strongly correlated with occupation. Therefore, additional adjustments for income would likely add limited information and could instead increase the risk of overfitting the regression model.

A minor issue is the statement in the introduction “Approximately 25% of IBD cases have a pediatric onset”. The ref. they cite took that statement from a review article. I think this is a myth that has been cited many times without a validated source/ref. based on a good research/population-based study. It originally came from a textbook from 1980s! Please revise.

- **Response:** Thank you for noticing this detail. We have now removed this statement from the introduction and revised the paragraph accordingly. Please see lines 53-61.

Some references were not in the journal format (e.g. Ref 27 lacking the year, ref 28)

- **Response:** The references have now been checked and corrected to ensure consistency with the journal's reference style.

Reviewer #2

This was a significant endeavour to collect 56,097 mother-child pairs of food frequency data and to have that number of offspring followed to identify pediatric-onset IBD cases. Their method to derive dietary patterns using an a posteriori approach rather than an a priori approach brings a unique angle to this type of research.

- **Response:** Thank you very much for these encouraging comments. We appreciate your recognition of the effort involved in assembling this large cohort and of our approach to derive data-driven dietary patterns.

A few points if the authors could elaborate or address, would strengthen the manuscript

1) is the one time food frequency questionnaire sufficient to reflect the dietary intake throughout pregnancy and into breastfeeding?

- **Response:** Maternal diet was assessed at gestational week 25, during the second trimester. Previous studies have shown that dietary patterns remain stable throughout pregnancy, despite some variation in specific food groups (Cucó G, et al. *Eur J Clin Nutr* 2006;60(3):364–71), supporting the use of a single assessment to capture habitual intake during this period. Still, we acknowledge that assessing diet only once is a limitation, as the impact of diet on offspring health may potentially vary across trimesters. We have now addressed this in the Discussion (lines 342-348). Regarding maternal diet during breastfeeding, this may differ from pregnancy; however, since our study specifically focused on the pregnancy period, it was outside the scope of the present work. However, we have now expanded the Discussion to include an explanation about the potential mediating effect of breastfeeding, please see lines 308-313.

2) delivery method - although possibly a controversial risk factor - there was no mention of the variable of delivery method being analyzed.

- **Response:** Delivery method could indeed be a risk factor for IBD due to its potential impact on offspring's immune and gut microbiota development. We have now expanded the discussion to include an explanation of this (line 290-297). Despite this, we did not adjust for delivery method in the primary analyses, as there is no evidence that it is associated with maternal diet during pregnancy, making confounding unlikely. To address your comment, we have instead included this variable in the descriptive Table 1, showing no clear differences between vaginal and cesarean deliveries across maternal dietary patterns.

3) they mention "breastfeeding duration" was assessed, but did not mention about formula feeding (which formula feeding or non-exclusive breastfeeding could affect offspring risk of developing IBD).

- **Response:** We acknowledge that postnatal feeding practices could influence offspring's IBD risk and, potentially, mediate part of the observed association between maternal dietary patterns during pregnancy and IBD risk in offspring. We have now expanded our Discussion section to include this (lines 342-348). While we assessed breastfeeding duration, detailed information on formula feeding was not available. However, we have instead added a variable describing the number of days of exclusive breastfeeding, with periods without exclusive breastfeeding assumed to involve formula or mixed feeding. Although we cannot distinguish between formula types, this addition provides partial information on postnatal feeding patterns and is included in Table 1.

Reviewer #3

The authors are to be commended on this well designed and well written study that I thoroughly enjoyed reading. Only one question: It does not appear that there was adjustment for child's antibiotics use in the first year of life, which is known to be associated with pediatric IBD. Did the authors look at this?

Minor:

- Typo in Figure 2 spider plots for sugar free soft drinks (says "sugerfree")
- The link in the data availability statement does not work (says "Page not found")
 - **Response:** Thank you, we are pleased to hear that you found the manuscript enjoyable to read. We have now corrected the typo and updated the link in the data availability statement. Regarding antibiotic use in the first year of life, we did not adjust for this in the analyses, as offspring antibiotic use is unlikely to confound the association between maternal diet and offspring IBD risk. However, to address your comment, we have now performed an additional analysis that further adjusted for the offspring's antibiotic use in the first year of life, which did not change the results (Supplementary Table 4).